# Customization and acceptability of the WHO labor care guide to improve labor monitoring among health workers in Uganda. An iterative development, mixed method study

**Godfrey R. Mugyenyi**[1]*, **Josaphat K. Byamugisha**[2], **Wilson Tumuhimbise**[3], **Esther C. Atukunda**[4], **Yarine T. Fajardo**[1]

1 Department of Obstetrics and Gynaecology, Mbarara University of Science and Technology, Mbarara, Uganda, 2 Department of Obstetrics and Gynaecology, Makerere University College of Health Sciences, Kampala, Uganda, 3 Department of Information Technology, Mbarara University of Science and Technology, Mbarara, Uganda, 4 Department of Pharmacy, Mbarara University of Science and Technology, Mbarara, Uganda

* gmugyenyi@must.ac.ug

**Data Availability Statement:** Availability of data and materials: A couple of de-identified transcripts have been provided alongside this submission. The

## Abstract

Cognisant of persistently high maternal and perinatal mortality rates, WHO called for adoption and evaluation of new adaptable and context-specific solutions to improve labor monitoring and health outcomes. We aimed at customizing/refining the new WHO labour care guide (LCG) to suite health care provider needs (HCP) in monitoring labour in Uganda. We used mixed methods to customize/refine and pilot test the new WHO LCG using stakeholder perspectives. Between 1st July 2023 and 30th December 2023, we conducted; 1)30 stakeholder interviews to identify user needs/challenges that informed initial modifications of the WHO LCG; 2)15 HCP in-depth interviews to identify any further needs to modify the LCG; 3) Two focus group discussions and 4) Two exit expert panels to identify any further user needs to further refine proposed modifications into the final prototype. Questionnaires were administered to assess acceptability. We interviewed 125 stakeholders with median age of 36 years (IQR;26–48) exposed to the LCG for at least 12 months with 11.8(SD = 4.6) years of clinical practice. Simple useful modifications/customizations based on format, HCP's perceived function and role in improving decision making during monitoring labour included; 1) Customizing LCG by adding key socio-demographic data; 2) Adjusting observation ordering; 3) Modification of medication dosages and 4) Provision for recording key clinical notes/labour outcome data on reverse side of the same A4 paper. All HCPs found the modified WHO LCG useful, easy to use, appropriate, comprehensive, appealing and would recommend it to others for labour monitoring. It was implementable and majority took less than 2 minutes to completely record/fill observations on the LCG after each labour assessment. Active involvement of end-users improved inclusiveness, ownership, acceptability and uptake. The modified LCG prototype was found to be simple, appropriate and easy-to-use. Further research to evaluate large-scale use, feasibility and effectiveness is warranted.

parent study is still ongoing and appropriate data sets will be available on request from the authors. Eventual data sets will be deposited in a repository for future reference.

**Funding:** The authors received no specific funding for this work.

**Competing interests:** The authors have declared that no competing interests exist.

## Introduction

Uganda's maternal mortality rate remains unacceptably high, and one of the highest in the world at 336 per 100,000 live births; perinatal mortality rates are 41/1000 live births [1]. To avert the persistently high maternal and perinatal deaths, scholars have discouraged use of ineffective practices in labor and childbirth [2]; WHO has called for adoption and evaluation of new adaptable and context-specific health solutions to improve labor monitoring and promote better health outcomes [3, 4]. There has been difficulty in drawing useful conclusions about costs, and adaptability of new approaches aimed at improving health outcomes because of complexity and methodologies used to integrate new health interventions [5, 6]. However, end-user centered iterative approaches that aim at characterizing, adopting, refining and integrating the new approaches into routine care have been observed to improve uptake and sustained use of these interventions among intended users [7].

For over 5 decades, a partograph has been the standard tool used to monitor labour globally. However, despite decades of training and investment in using this tool for labour monitoring, the rates of maternal-perinatal outcomes remain poor in many countries. Some scholars have argued that the partogram majorly offers subjective variations, and assumes that all women progress at same rate, which may affect intervention rate and affect health outcomes for both mother and baby [8]. This subjective nature, coupled with the reportedly low acceptability of this partograph have grossly affected effective use of the partograph to monitor labour among others [8, 9]. Cognisant of the persistent maternal and perinatal mortality rates, the World Health Organization (WHO) has made new recommendations on intrapartum care to improve labour monitoring as well as maternal childbirth experience [4, 10]. The WHO has recommended that the partograph, a tool traditionally used to monitor labour globally, be modified to fit emerging user needs, available evidence and global priorities aimed at reducing the persistently high maternal-perinatal mortalities and morbidities [11–13].

WHO guidelines and recommendations included new and updated definitions and considerations such as duration of the first and second stages of labour which was recognised to be variable between women, cervical dilatation rate as a sole and poor predictor of obstetric intervention and birth outcomes, making the partograph scientifically invalid [4, 10, 14, 15]. The WHO emphasized the importance of incorporating woman-centred care within the monitoring tool to optimize the experience of labour and childbirth for women and their babies. Based on the available evidence, a new labour care guide (LCG; Fig 1) was designed as a "next-generation" partograph to help health care providers (HCP) effectively monitor the well-being of women and babies during labour through regular assessments and reporting [4]. Unlike the traditional partograph, the new LCG tool aims to stimulate shared decision-making between HCPs and women in order to promote women-centered care. The new LCG also aims to continuously remind practitioners to offer supportive care throughout labour and childbirth, and remind them of what observations need to be regularly made during labour to identify any emerging deviations from normal or complications in mother and/or baby. Unlike the current partograph design, the LCG provides an avenue to monitor important supportive care interventions such as labour companionship, birth position, pain relief, women's mobility, known to improve childbirth experience and health outcomes. According to WHO, this LCG further provides new reference thresholds and alerts for abnormal labour observations which are meant to trigger specific actions [16]. In adding all these components, the adoption of this tool as a labour monitoring-to-decision tool is expected/hypothesized by the clinical trial underway, to enhance quality of care during labour and birth, support prompt audits and minimize over-diagnosis and under-diagnosis of abnormal labour events, reduce use of unnecessary interventions such as caesarean sections and or labour augmentation [16]. An open-label

## WHO LABOUR CARE GUIDE

Name Parity Labour onset Active labour diagnosis [Date ]

Ruptured membranes [Date Time ] Risk factors

| | | ALERT | | ACTIVE FIRST STAGE | | | | | | | | | | | | SECOND STAGE | | |
|---|---|---|---|---|---|---|---|---|---|---|---|---|---|---|---|---|---|---|
| | Time (:) | | | | | | | | | | | | | | | | | |
| | Hours | | 1 | 2 | 3 | 4 | 5 | 6 | 7 | 8 | 9 | 10 | 11 | 12 | 1 | 2 | 3 |
| **SUPPORTIVE CARE** | Companion | N | | | | | | | | | | | | | | | | |
| | Pain relief | N | | | | | | | | | | | | | | | | |
| | Oral fluid | N | | | | | | | | | | | | | | | | |
| | Posture | SP | | | | | | | | | | | | | | | | |
| **BABY** | Baseline FHR | <110, ≥160 | | | | | | | | | | | | | | | | |
| | FHR deceleration | L | | | | | | | | | | | | | | | | |
| | Amniotic fluid | M+++, B | | | | | | | | | | | | | | | | |
| | Fetal position | P, T | | | | | | | | | | | | | | | | |
| | Caput | +++ | | | | | | | | | | | | | | | | |
| | Moulding | +++ | | | | | | | | | | | | | | | | |
| **WOMAN** | Pulse | <60, ≥120 | | | | | | | | | | | | | | | | |
| | Systolic BP | <80, ≥140 | | | | | | | | | | | | | | | | |
| | Diastolic BP | ≥90 | | | | | | | | | | | | | | | | |
| | Temperature ºC | <35.0, ≥37.5 | | | | | | | | | | | | | | | | |
| | Urine | P++, A++ | | | | | | | | | | | | | | | | |
| **LABOUR PROGRESS** | Contractions per 10 min | ≤2, >5 | | | | | | | | | | | | | | | | |
| | Duration of contractions | <20, >60 | | | | | | | | | | | | | | | | |
| | Cervix [Plot X] 10 | | | | | | | | | | | | | | | | | |
| | 9 | ≥ 2h | | | | | | | | | | | | | | | | |
| | 8 | ≥ 2.5h | | | | | | | | | | | | | | | | |
| | 7 | ≥ 3h | | | | | | | | | | | | | | | | |
| | 6 | ≥ 5h | | | | | | | | | | | | | | | | |
| | 5 | ≥ 6h | | | | | | | | | | | | | | | | |
| | Descent [Plot O] 5 | | | | | | | | | | | | | | | | | |
| | 4 | | | | | | | | | | | | | | | | | |
| | 3 | | | | | | | | | | | | | | | | | |
| | 2 | | | | | | | | | | | | | | | | | |
| | 1 | | | | | | | | | | | | | | | | | |
| | 0 | | | | | | | | | | | | | | | | | |
| **MEDICATION** | Oxytocin (U/L, drops/min) | | | | | | | | | | | | | | | | | |
| | Medicine | | | | | | | | | | | | | | | | | |
| | IV fluids | | | | | | | | | | | | | | | | | |
| **SHARED DECISION-MAKING** | ASSESSMENT | | | | | | | | | | | | | | | | | |
| | PLAN | | | | | | | | | | | | | | | | | |
| | INITIALS | | | | | | | | | | | | | | | | | |

In active first stage, plot 'X' to record cervical dilatation. Alert triggered when lag time for current cervical dilatation is exceeded with no progress. In second stage, insert 'P' to indicate when pushing begins.

INSTRUCTIONS: CIRCLE ANY OBSERVATION MEETING THE CRITERIA IN THE 'ALERT' COLUMN, ALERT THE SENIOR MIDWIFE OR DOCTOR AND RECORD THE ASSESSMENT AND ACTION TAKEN. IF LABOUR EXTENDS BEYOND 12H, PLEASE CONTINUE ON A NEW LABOUR CARE GUIDE.

Abbreviations: **Y** – Yes, **N** – No, **D** – Declined, **U** – Unknown, **SP** – Supine, **MO** – Mobile, **E** – Early, **L** – Late, **V** – Variable, **I** – Intact, **C** – Clear, **M** – Meconium, **B** – Blood, **A** – Anterior, **P** – Posterior, **T** – Transverse, **P+** – Protein, **A+** – Acetone

*(left margin, vertical)* WHO LABOUR CARE GUIDE: USER'S MANUAL

**Fig 1. The new WHO labour care guide.**

randomized controlled trial on 280 low-risk antenatal women admitted for delivery at a busy tertiary care institute in North India showed significant reduction in Caesarean delivery rates among women monitored using a WHO LCG (1.5%) compared to control group (17.8%; $P$ = .0001) [17]. The duration of active phase of labour was also shortened significantly in the study group, with acceptability and satisfaction levels reported to be high in LCG use. A total of 136 doctors, midwives, and nurses in twelve health facilities across Kenya, Tanzania, India, Nigeria, Malawi and Argentina who applied the LCG in managing labour and childbirth of 1,226 low-risk women reported high satisfaction, usability and acceptability levels [18].

Identifying and scaling up such context-specific interventions have great potential to improve quality of healthcare and outcomes in pregnancy, childbirth and immediate postnatal period [13, 19–21]. The Labour Care Guide manual has been developed by WHO to help skilled health personnel to successfully use the tool. However, the tool has not been adapted and adopted to be used as a decision-making tool in any low- and middle-income country maternity care settings, with correspondingly high rates of prolonged/obstructed labor, unnecessary interventions, maternal and perinatal mortality and morbidity. Whereas the barriers and challenges of using partographs are widely documented [22], there is little documented effect of the new LCG, and the entire implementation strategy has not been evaluated in low resource settings. There is also an urgent need for not only adoption studies, but also refining and testing an effective implementation strategy that not only addresses the current contextual barriers to labour monitoring, but also fosters sustained uptake and use of the new LCG as a recommended standard for providing and improving routine intrapartum care globally. In this study, we attempt to customize, refine and test the new WHO labour care guide among health care providers monitoring labour in South Western Uganda, as we explore a suitable/appropriate implementation strategy for the new tool in a Ugandan context.

## Methods

### Study design

Using mixed methods and iterative development design described by Mugyenyi GR et al 2024 [23], we aimed at customizing, refining and pilot testing a user-centered LCG tool that would be comprehensive and easy-to-use, so as to optimize outcomes and long-term use by HCPs while monitoring labour in Uganda (Fig 2). Using an interview guide, we carried out 30

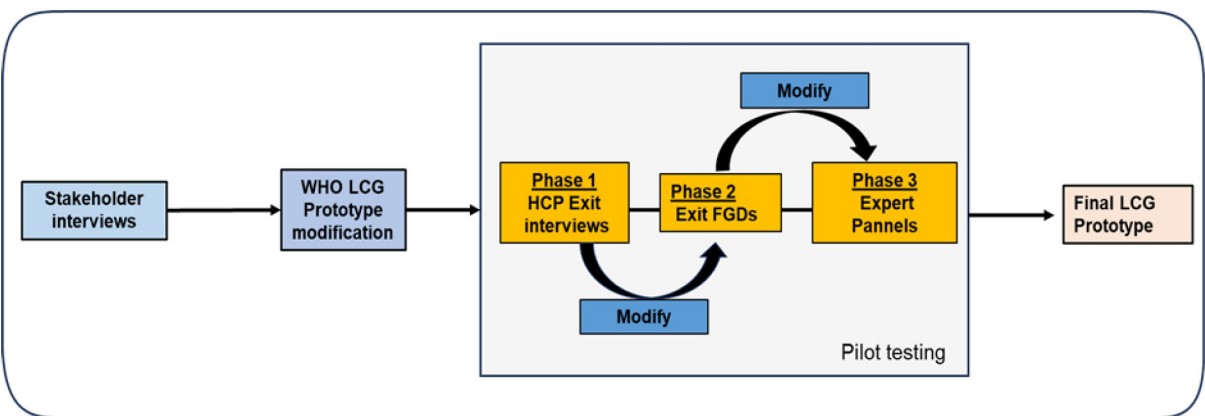

**Fig 2. Modified WHO labour care guide development process.**

stakeholder interviews (see stakeholder interview section below for details) to identify facilitators of continuous labour monitoring in Uganda, current user needs, and perceived LCG tool opportunities, complications or challenges (formative phase). This feedback informed the first modification of the WHO LCG. We then deployed this modified LCG (prototype one) for use among all HCPs at Mbarara Regional Referral Hospital (MRRH), a referral hospital with the biggest maternity center, with over 90 HCPs involved in labour monitoring in Southwestern Uganda. After 2 months of use, we purposively conducted a subset of 15 HCP exit in-depth interviews, including HCPs across all cadres (at least 3 for each cadre) to identify any further needs, gaps, experiences and challenges. This information was analyzed and utilized to make simple suggested modifications, that resulted into prototype two. The new modified LCG was then re-deployed for further use by all HCPs across three Mbarara city public facilities; at least one Health center IV and 2 Health center IIIs. After the 2 months of deployment, we conducted two Focus Group Discussions (FGDs) of 10 participants each involving midwives, interns, residents and consultants from these three facilities to identify any further user needs, gaps and or challenges. This prototype was then deployed for use for at least 2 months by all HCPs at MRRH. Questionnaires were then administered to assess acceptability and usability of the modified LCG. Two expert panels involving 4 consultant obstetricians, 4 Senior House Officers, 2 Medical Officers and 10 midwives that had used this tool for at least two months during this study were constituted to further refine and consolidate all proposed modifications and components of LCG into the final prototype, ready for evaluation (Figs 3 and 4). These steps were aimed at refining and customizing a user-centered tool for easy uptake and use within an established intervention development framework [24–26]. Data collection took place between 1st of July 2023 and 30th December 2023.

## Study setting

Mbarara City just like elsewhere in Uganda has a public health system that is organized in five tiers, with a regional referral hospital and four levels of community health centers. Staffing and available maternity services vary across the four levels: HCIII also referred to as basic emergency obstetric and newborn care (BeMONC) facilities carry out vaginal deliveries, whereas HCI and HCII serve as low resource community referral units. HCIVs and hospitals, referred to as comprehensive emergency obstetric and newborn care (CeMONC) facilities conduct caesarean deliveries and offer blood transfusion services [1]. Several private facilities operate in parallel to the public health system to provide maternal healthcare. There is one regional referral hospital (MRRH) that serves the Southwestern region, where most of deliveries are high-risk [27], with 14 Obstetricians and Gynaecologists, 31 Residents, 10 Intern doctors, and 38 Midwives. The nurse-to-patient ratio stands at about 1:25 [28]. The department performs about 10,000 deliveries annually, majority of whom deliver vaginally [29]. Women who deliver from the department are usually transferred from the antenatal clinic, referred from peripheral health units, or come directly from the community. Although often incompletely implemented, with suboptimal utilization and labour outcomes, all mothers in labour remain monitored using a partogram, a graph of labour parameters and cervical dilatation over time, with pre-printed alert and action lines designed to prompt intervention if a woman's curve deviates from the expected course [30–33]. The foetal heart rate in Uganda is monitored manually per clinician judgment using the Pinard's stethoscope. Only one electronic monitor (CTG machine) and one ultrasound scan are available for use in labour, and are occasionally used to confirm presence or absence of regular cardiac pulsations, but not for monitoring labour [34]. Mbarara City is located ~270 kilometres Southwest of the capital, Kampala [35].

# MINISTRY OF HEALTH
## MODIFIED WHO LABOUR CARE GUIDE

Hospital or Health Centre: _________________________________________ **IP**No: _______________________

Name: ___________________________________________Age: _________ NIN: _______________________________

Parity: _______LNMP [Date___/___/___] Weeks of Gestation _______ Labour onset [Date___/____/____, Time ___:____]

Active labour diagnosis [Date____/____/___Time: ____: ___] Rupturedmembranes [Date___ /____ /____,Time___: ____]

Risk factors: ___________________________________________________________________________

PMTCT code ___________________ Syphilis: Positive ☐ Negative ☐ Hepatitis B: Positive ☐ Negative ☐

| | | ALERT COLUMN | Time | | | | | | | | | | | | | | 123 | | |
|---|---|---|---|---|---|---|---|---|---|---|---|---|---|---|---|---|---|---|---|
| | | | Hours | 1 | 2 | 3 | 4 | 5 | 6 | 7 | 8 | 9 | 10 | 11 | 12 | | | |
| | | | ALERT | ◄ | | | | ACTIVE FIRST STAGE | | | | | | | ► | ◄ SECOND STAGE ► | | |
| SUPPORTIVE CARE | | Companion | N | | | | | | | | | | | | | | | | |
| | | Pain Relief | N | | | | | | | | | | | | | | | | |
| | | Oral fluid | N | | | | | | | | | | | | | | | | |
| | | Posture | SP | | | | | | | | | | | | | | | | |
| BABY | | Baseline FHR | <120, ≥160 | | | | | | | | | | | | | | | | |
| | | FHR decelerations | L | | | | | | | | | | | | | | | | |
| | | Amniotic fluid | M+++, B | | | | | | | | | | | | | | | | |
| | | Fetal position | P,T | | | | | | | | | | | | | | | | |
| | | Caput | +++ | | | | | | | | | | | | | | | | |
| | | Moulding | +++ | | | | | | | | | | | | | | | | |
| WOMAN | | Pulse | <60,≥120 | | | | | | | | | | | | | | | | |
| | | Systolic BP | <80, ≥140 | | | | | | | | | | | | | | | | |
| | | Diastolic BP | ≥90 | | | | | | | | | | | | | | | | |
| | | Temperature °C | <35.0, ≥37.5 | | | | | | | | | | | | | | | | |
| | | Urine | P++, A++ | | | | | | | | | | | | | | | | |
| LABOUR PROGRESS | | Contractions per 10min | ≤2, >5 | | | | | | | | | | | | | | | | |
| | | Duration of contractions | <20, >60 | | | | | | | | | | | | | | | | |
| | Cervix [Plot X] | 10 | | | | | | | | | | | | | | | | | |
| | | 9 | ≥2h | | | | | | | | | | | | | | | | |
| | | 8 | ≥2.5h | | | | | | | | | | | | | | | | |
| | | 7 | ≥3h | | | | | | | | | | | | | | | | |
| | | 6 | ≥5h | | | | | | | | | | | | | | | | |
| | | 5 | ≥6h | | | | | | | | | | | | | | | | |
| | Descent [Plot 0] | 5 | | | | | | | | | | | | | | | | | |
| | | 4 | | | | | | | | | | | | | | | | | |
| | | 3 | | | | | | | | | | | | | | | | | |
| | | 2 | | | | | | | | | | | | | | | | | |
| | | 1 | | | | | | | | | | | | | | | | | |
| | | 0 | | | | | | | | | | | | | | | | | |

*In active first stage plot "X" to record cervical dilatation. Alert triggered when lag time for current cervical dilatation is exceeded with no progress. In second stage, insert "P" to indicate when pushing begins.*

| SHARED DECISION-MAKING | IMPRESSION / ASSESSMENT | | | | | | | | | | | | | | | | |
|---|---|---|---|---|---|---|---|---|---|---|---|---|---|---|---|---|---|
| | ACTION PLAN | | | | | | | | | | | | | | | | |
| MEDICATIONS | Oxytocin | Dose (IU IN 500MLS) | | | | | | | | | | | | | | | |
| | | Rate (Drops/min) | | | | | | | | | | | | | | | |
| | IV fluids | Type | | | | | | | | | | | | | | | |
| | | Volume | | | | | | | | | | | | | | | |
| | INITIALS | | | | | | | | | | | | | | | | |

**INSTRUCTIONS:**
1. CIRCLE ANY OBSERVATION MEETING THE CRITERIA IN THE 'ALERT' COLUMN.
2. ALERT THE SENIOR MIDWIFE OR DOCTOR AND RECORD THE ASSESSMENT AND ACTION TAKEN.
3. IF LABOUR EXTENDS BEYOND 12H, PLEASE CONTINUE ON A NEW LABOUR CARE GUIDE.

**Abbreviations:**Y – Yes, N – No, D – Declined, U – Unknown, SP – Supine, MO – Mobile, E – Early, L – Late, V – Variable, I – Intact, C – Clear, M – Meconium, B – Blood, A – Anterior, P – Posterior, T – Transverse, P+ – Protein, A+ – Acetone

**Fig 3. Modified labour care guide for Uganda (front page).**

**Number of ANC visits** [____] **1st ANC visit** [Date___/___/____] **Haemoglobin**(Hb)___g/dl [Date___/___/_____]

**Additional clinical notes:** ________________________________________________

__________________________________________________________________________

__________________________________________________________________________

__________________________________________________________________________

__________________________________________________________________________

## OUTCOME OF LABOUR

**MOTHER**

Date of delivery _________________Time of Delivery: ___________ Type of delivery: ________________________

Duration 1st Stage: __________Duration of 2nd Stage____________Amount of Blood Loss (Mls) ________________

**PLACENTA AND MEMBRANES:** Complete; Yes [ ] No [ ]     Abnormalities: Yes [ ] No [ ]

 If Yes, specify________________________________________________________________

**MEDICINES GIVEN**

| Name | Route | Dose | Other Medicines Given | Route | Dose |
|------|-------|------|----------------------|-------|------|
| Oxytocin | | | | | |
| Misoprostol | | | | | |
| Cabetocin | | | | | |
| Tranexamic acid | | | | | |
| Other (specify) | | | | | |

**GENITAL TRACT/PERINEUM**

Episiotomy performed: Yes [ ] No [ ]     Tears: Yes [ ] No [ ]     Repaired: Yes [ ] No [ ]

**BABY**

Alive    Yes [ ] No [ ]  Apgar Score (1 min) __________________(5 minutes) ____________________

Sex__________Weight _______ (Kgs).    Abnormalities: Yes [ ] No [ ]   If Yes, specify___________________

Tetracycline given: Yes [ ] No [ ]    Cord Care: Yes [ ] No [ ]

Vit K given: Yes [ ] No [ ] (Dose is 1 mg Term Baby/0.5mg – premature)

Nevirapine (eMTCT): Yes [ ] No [ ]

*Dosing for 10mg/ml formulation: Birth weight 2.0 - 2.5kg - 1m once daily, Birth weight >2.5kg – 1.5ml once daily*

**Delivered by:** ________________________**Cadre**____________ **Assisted by** __________________

Post Delivery BP: _______________ Post delivery Pulse: __________________ Temperature: __________________

Immediate Postpartum Care (0-1 hours) including any treatment given:

__________________________________________________________________________

__________________________________________________________________________

__________________________________________________________________________

__________________________________________________________________________

Breast feeding initiated within 1 hour Yes [ ]    No [ ]   New-born Temperature _______⁰C

**Immunizations Given:** BCG: Yes    No    Polio 0: Yes    No    Hepatitis B: Yes    No

**Fig 4. Modified labour care guide for Uganda (back page).**

**Tool refinement and pilot testing.** We utilized Behavioral Change Taxonomy [36, 37] to identify, refine/modify and customize key LCG components needed to optimize knowledge of danger signs, cues, prompts, timely plan action, and maximize its long-term use and impact among HCP LCG users. We also used credible source of information from the intended HCP users, plus experts to customize information and instructions needed for self-monitoring, perform the needed behavior (complete LCG during monitoring) and improve HCP interaction for support and problem solving in our setting.

## Stakeholder interviews

We enrolled and interviewed 30 HCPs and Ministry of Health/WHO officials between July and September 2023 to achieve saturation of data. HCPs were purposively selected from 4 facilities across Mbarara City, South Western Uganda to represent diverse labour monitoring experiences and ideas about the new WHO LCG. We also explored challenges and opportunities of labour monitoring in the present times. We included other specialist HCPs and MoH/WHO officials exposed to the new WHO LCG. The data from these interviews was used to identify key LCG components, gaps, information that would be used to help customize the new WHO LCG into a useable version (prototype one) for HCPs in Uganda. The detailed data from these stakeholder interviews were analyzed using the Consolidated Framework for Implementation Research (CFIR) and have been published elsewhere [38].

**HCP exit interviews.** A total of 15 exit HCP interviews were conducted following use of the modified LCG (prototype 1) for at least 2 months at MRRH. Participants of different cadre and qualifications were screened and purposively selected from MRRH to explore user experiences, opportunities, gaps and ideas until saturation, to improve the prototype. This data was synthesized and used to further refine the LCG into prototype two. This modified version (prototype two) was re-deployed in three Mbarara City facilities for use and testing in one health center IV and two health center IIIs. All interviews were open-ended and covered a wide range of topics following the interview guide developed to explore ease of use, perceived usefulness, the HCP attitude to LCG use, preference, effort and performance expectancy, as well as other useful topics like self-efficacy, behavioral intention to use and actual use of the modified WHO LCG. A brief data on demographic information (e.g., age, qualification, work experience) was collected. All interviews were conducted in a private location agreed upon by the HCPs. All interviews lasted 40–60 minutes, and written informed consent was obtained from all participants at the start of each interview session. All interviews were recorded digitally, with participant's permission and transcribed verbatim (see S1 Text).

**Focus group discussions.** Two FGDs were done to include 10 HCPs each of all cadres that utilized prototype two to monitor labour for at least two months in the three public facilities discussed above. These FGDs were conducted to explore experiences, gaps and obtain feedback on the modified prototype one. This input/feedback was synthesized, and ranked to refine components of a subsequent prototype two.

**Final LCG prototype testing.** The refined prototype two was deployed for use in routine labour monitoring by all HCPs at MRRH, the busiest facility in South Western Uganda for at least 2 months. At the end of 2 months of use, a self-administered structured questionnaire was used to obtain demographic information, as well as feedback on appeal, ease of use, complexity, content, usefulness, tool's preference, time taken to plot, fill and complete tool after assessment, its appropriateness and feasibility in this setting.

### Expert pannels

This prototype two was further subjected to two expert panels each comprising of 10 HCPs of different cadres (2 certificate midwives, 8 diploma midwives, 6 bachelor's degree nurses, medical officers and residents, and 4 consultant obstetricians) to provide feedback on specific LCG prototype two components, plus experiences of this version's functionality and ease-of-use to monitor labour in Uganda. Following these two expert panels, we further refined and consolidated all proposed components of the modified LCG to produce the final LCG (prototype three) for Uganda, ready for evaluation (Figs 3 and 4). This final prototype will be evaluated in effectively monitoring labour and improving clinical outcomes among all HCPs in Mbarara district and Mbarara City, South Western Uganda as the next step.

All baseline participant characteristics were collected from all study participants. Questionnaire data collection and in-depth interviews were performed in English. Two trained research assistants conducted the interviews. A facility inventory was done to inform our understanding of baseline conditions and set up for maternity and labour monitoring services at MRRH, Health Center IVs, and Health Center IIIs.

### Data analysis

We described demographic and clinical data for all interviewed participants using standard descriptive statistics. Qualitative analysis included repeated review of transcripts to identify relevant experiences, ideas and preferences on labour monitoring and LCG use. We coded qualitative data using NVivo (version 12.0; Melbourne, Australia), and iteratively reviewed and sorted codes to identify repeated themes using inductive content analysis [39]. We used illustrative quotes taken from the qualitative interviews to illustrate the context and meaning in each theme. Data analysis was done jointly by GRM and WT to ensure consistency in coding.

**Ethical considerations.**   Ethical clearance was obtained from the Faculty Research Committee in the Faculty of Medicine and the Research Ethics Committee (REC) at Mbarara University of Science and Technology (MUST-2023-808). Study site administrative clearance was obtained from the Hospital Director of Mbarara Regional Referral Hospital and Mbarara City Health Officer for Mbarara City. We sought approval from the National Council for Science and Technology in Uganda (UNCST)–HS2864ES. We obtained written informed consent from all study participants before data collection started.

**Ethical compliance with human study.**   This study was conducted in compliance with the ethical standards of the responsible institution on human subjects as well as with the Helsinki Declaration.

## Results

We interviewed a total of 125 participants for this development phase. The median age of the interviewed HCPs was 36 years (IQR;26–48), distributed across enrolled midwives (14%) diploma midwives (37%), bachelors/Residents (31%) and consultants (18%). The mean years of practice for the HCPs was 11.8 (SD = 4.6), and all HCPs had at least 12 months of exposure to the LCG (Table 1).

### LCG design, refinement and update

From all qualitative interviews, a number of simple modifications and inclusions were made on the original WHO LCG tool (Fig 1), based on format and HCP's perceived function, role, or goal in monitoring labour and improving decision making or action planning in Uganda and similar settings. HCPs identified and defined prototype design and content needs that

**Table 1. Demographic characteristics of study participants.**

| Characteristic | Stakeholder Interviews (n = 30) | HCP interviews (n = 15) | FGD Interviews (n = 20) | Pilot test (n = 40) | Expert Panels (n = 20) | All Participants (n = 125) |
|---|---|---|---|---|---|---|
| Median age (IQR) | 36 (27,54) | 38(28, 45) | 31(25,47) | 35(25,52) | 36(26,48) | 36(26,48) |
| Mean duration of clinical practice in years (SD) | 11.3 (7.7) | 15.2 (4.7) | 8(5.2) | 10.2(4.4) | 9.2(5.5) | 11.8(4.6) |
| HCP Cadre, n (%) | | | | | | |
| Certificate Midwife | 5(13.9) | 3(20) | 2(10) | 6(15) | 2(10) | 18(14.4) |
| Diploma Midwife* | 10(27.8) | 3(20) | 8(40) | 17(42.5) | 8(40) | 46(36.8) |
| Bachelors/Residents* | 10(36.1) | 6(40) | 6(30) | 11(27.5) | 6(30) | 39(31.2) |
| Consultant* | 5(13.9) | 3(20) | 4(20) | 6(15) | 4(20) | 22(17.6) |
| Mean time exposure to LCG (months) | 12.3(4.2) | 12.0(3.8) | 13(2.5) | 12.0(2.5) | 12.4(2.5) | 12.3(2.8) |

were expected to help them optimize LCG long-term use, and rates of completeness with its intended benefits. These useful customizations, also summarized in Table 2, included; 1) Customizing LCG by adding key socio-demographic data compatible with existing programs to aid planning and managing risk; 2) Adjusting observation ordering to facilitate an easy-to-use interface, flow, familiarization, engagement and clarity; 3) Modification of key record and dosage to suit local context; 4) provision for a section to capture key clinical notes and labour outcome data to facilitate auditing, accountability, reference, utilization and immediate postpartum care on the reverse side of the LCG A4 paper.

**1) Customizing LCG by adding key socio-demographic data compatible with existing programs.** HCPs reported that there were key socio-demographic data that was missing in the original WHO LCG that needed to be added in order to aid planning care and managing risk among women enrolled in labour. These included the Hospital/facility name, in-patient number (IPN), patient age and National Identification number (NIN), for easy patient identification and inter-facility tracking in times of patient transfer/referral. These are currently routine in all public facilities in Uganda. Other suggested additions included; Last normal menstrual period (LNMP), weeks of gestation, with specific dates and time format of other events like labour onset, active labour diagnosis and rapture of membranes, prevention of mother to child transmission (PMTCT) code, Syphilis and Hepatitis B test results, aimed at screening for HIV/AIDS, syphilis and Hepatitis B respectively for all mothers in labour (also known as triple elimination). These according to the HCPs, could help them to improve uniformity and consistency in documenting labour events, make the new LCG compatible with the existing programs, as well as make planning and management of risk easier and better when included on the same form. Interviewed HCPs also noted that this suggested information was already part of MOH's existing routine labour monitoring parameters/program that triggers any HCPs to spontaneously assess and clarify the categories of women at risk and in need of close monitoring during labour, and that it would not require a lot of additional training to use and fill the LCG that uses familiar indices. According to a Senior Health Officer, with 12 years of experience in labour monitoring,

*"Each mother is unique and so is their management plan. A lot of this [triple elimination] information gives the managing team a good perspective on the nature of risk or mother you are managing. It is a good way to quickly translate information to another team taking over for example, the women who are at risk of infection such as HIV and others as per the existing MOH policies and practices".*

**Table 2. Partograph, WHO labour care guide and the modified LCG.**

**Differences between the modified partograph, who LCG and the modified LCG**

| Parameter | Modified WHO partograph | WHO Labour Care Guide | Modified Labour Care Guide Prototype 3 |
|---|---|---|---|
| Name of Health Center | No | No | Yes |
| In Patient Number | No | No | Yes |
| Name of Patient | Yes | Yes | Yes |
| Age | No | No | Yes |
| National Identification Number (NIN) | No | No | Yes |
| Woman's gravidity | Yes | No | No |
| Woman's Parity | No | Yes | Yes |
| LNMP | No | No | Yes |
| Weeks of gestation | No | No | Yes |
| Date and time of admission | No | Yes | Yes |
| Time of rupture of membranes | Yes | Yes | Yes |
| Type of labour onset (spontaneous or induced) | No | Yes | Yes |
| Medical and social risk factors | No | Yes | Yes |
| HIV/Syphilis and Hepatitis Test results | No | No | Yes |
| Supportive care interventions (labour companionship, pain relief, oral fluid intake, and maternal position) | No | Yes | Yes |
| Fetal heart rate (FHR) | Yes * | Yes** | Yes** |
| Presence of early, late, or variable decelerations | No | Yes | Yes |
| Normal lower limit of FHR | 110 | 110 | 120 |
| Amniotic fluid characteristics | Yes | Yes | Yes |
| Fetal position | No | Yes | Yes |
| Caput | No | Yes | Yes |
| Moulding | Yes | Yes | Yes |
| Woman's pulse and blood pressure (BP) | Yes* | Yes** | Yes** |
| Woman's temperature | Yes | Yes | Yes |
| Urine volume | Yes | No | No |
| Proteinuria and acetonuria | Yes | Yes | Yes |
| Duration and frequency of uterine contractions | Yes | Yes | Yes |
| Strength of contractions | Yes | No | No |
| Definition of active phase | Starting from 4 cm of cervical dilatation | Starting from 5 cm of cervical dilatation | Starting from 5 cm of cervical dilatation |
| Definition of "satisfactory" labour progress | Fixed 1 cm/hour time limit ("alert" and "action" lines) | Evidence-based time limits at each centimeter*** | Evidence-based time limits at each centimeter*** |
| Cervical dilatation | Yes | Yes | Yes |
| Descent of the fetal head | Yes | Yes | Yes |
| Values for "normal" | "Alert" and "action" lines for cervical dilatation; thick lines to identify parameters for normal FHR | "Reference threshold" values*** are listed for non-clinical and clinical parameters | "Reference threshold" values*** are listed for non-clinical and clinical parameters |
| Second stage section | No | Yes (all parameters except cervical dilatation) | Yes (all parameters except cervical dilatation) |
| Time when pushing begins | No | Yes | Yes |

(*Continued*)

**Table 2.** (Continued)

| Differences between the modified partograph, who LCG and the modified LCG | | | |
|---|---|---|---|
| Parameter | Modified WHO partograph | WHO Labour Care Guide | Modified Labour Care Guide Prototype 3 |
| Identification of deviations from expected observations | No explicit way to document deviations from expected observations of any labour parameter, other than cervical dilatation to the right of "alert" and action lines and FHR 180 bpm or faster/100 bpm or slower | Requires circling any observations meeting the criteria in the "alert" column | Requires circling any observations meeting the criteria in the "alert" column |
| Assessment of findings | No | Yes | Yes |
| Plan of care | No | Yes | Yes |
| Provider's initials | No | Yes | Yes |
| Record of labour outcomes and other important labour and post-partum observations and management on the flip side of the paper | No | No | Yes |

*Values are plotted on a graph

**Values are written in the appropriate cell

***Reference threshold values for labour observations define normal, expected ranges for the different parameters. They are intended to trigger reflection and specific action(s) if an abnormal observation is identified.

*"It is easy to plan for this woman holistically when you have all the information you need in one place. . .it has helped a lot and I think we should maintain that information, and consistency especially since we don't need more training to assess the same things on the LCG. I feel like everyone is able to identify women quickly and draw a plan together on how best to support or transfer for better management",*

added a Senior Midwife, with 20 years of experience in labor monitoring.

**2) Adjusting observation ordering to facilitate an easy-to-use interface.** All HCPs suggested that the LCG sequence of observations needed to be re-aligned to be in synch with the flow of activities and sequence of normal and routine labour process. For example, the HCPs noted that; 1) Assessment section of the "shared decision making" could be modified to include impression/ assessment. The word "impression" as per the HCP's preference referred to diagnosis following examination and clinical assessment as was commonly interpreted and familiarized locally, 2) Under shared decision making, "plan" was suggested to be modified to "action plan" to prompt the attending clinician to make or document a suitable action plan, following complete assessment and interpretation of all labour observations, including alerts, 3) Section 6 (Medication) was suggested to come after shared decision-making (originally as section 7). This switch according to the HCPs was because medications routinely get prescribed following complete assessment and documentation of action plan preceding the assessment/ impression or diagnosis. All these modifications (as seen in Figs 3 and 4) according to the interviewed HCPs have potential to improve the tool's user interface, facilitate familiarization, logic flow of information, organization, HCP engagement and clarity during labour monitoring. One of the midwives who has had over 10 years' experience in managing labour said,

*"You see, we work in busy units where time is of essence. It is very helpful to arrange these observations in a manner that makes sense for most of us, but also in a way that helps us to*

*quickly document, organize and make sense of these observations easily and plan with others".*

A Senior Medical Officer with 12 years of experience managing labour also said,

*"When we get all this information available and ordered step-wise, and by way of routine practice, it's familiar and easier to communicate to others on the team. . .it flows well and everyone appreciates and works with the information better".*

**3) Modification of key record and dosage to suit local context.** HCPs interviewed during expert panel interviews noted that the lower limit of the baseline fetal heart rate needed to be modified from ≤110 to ≤120 beats per minute, as previously captured on the old partogram. This suggestion according to the HCPs could provide a safety margin in case of routine patient transfer or referral from the basic to the comprehensive emergency obstetric and neonatal care units that are usually far apart, and often with limited transportation/ambulance system. However, in an attempt to create space, some HCPs reported that fetal heart rate deceleration record was redundant and not applicable for most public facilities especially since it required CTG machines that are unavailable, and not mandatory in most Ugandan basic and comprehensive emergency obstetric care facilities, except for some regional referral and private hospitals. According to the HCPs expert panel review, this section should be maintained for the benefit of the facilities that have the capacity to acquire CTG machines, and for future plans and developments in obstetric care and management.

The interviewed HCPs reported that the oxytocin row under the medication section should be modified to accommodate the oxytocin dose in International Units (IU) administered in a 500mL unit of the available crystalloids (normal saline or ringers' lactate) locally in Uganda. The 1 litre unit provided for in the WHO LCG was reported to not be available on market in Uganda. According to the HCPs, an additional row to accommodate the rate of oxytocin administered at a particular time in drops per minute should be added to aid detailed documentation and interpretation of augmented labour progress. The hourly boxes should be bisected to accommodate the dose and rate for every 30 minutes of incremental oxytocin administration. Similar columns and rows should be added to accommodate the type and volume of intravenous fluid (IV fluids) used in labour. The separation according to interviewed HCPs would make the section easy to record the different fluids and volume during labour at a particular time, and make it less squeezed in one small box provided for by the original WHO LCG.

According to HCPs, the rows for capturing other medications, other than oxytocin used during active labour were redundant during active labour monitoring. The HCPs noted that any other medicines (not used for labour monitoring) including those given for co-existing infection, severe preeclampsia/eclampsia, epidural, tranexamic acid, oxytocin/ and other uterotonics given during active management of third stage of labour should be removed from the active labour monitoring section six, and recorded on the reverse side of the modified LCG to save more space and make the LCG less crowded. A senior 54-year-old midwife argued,

*"The LCG indicates the 1 litre intravenous bottles, which are not available in Uganda and so this recommendation would be redundant and not appropriate here [in Uganda] . . .Secondly, the medications other than oxytocin that is used during labour monitoring are not usually*

*part of labour monitoring and could be removed from the chart and recorded elsewhere to avoid confusion and save some space".*

*"I also noted that the basic maternity service centres cannot do C-sections, and often times need a good transfer window to transfer women with foetal heart of at least less than 120 beats per minute elsewhere for help. Moreover, I think in our setting with the kind of poor ambulance system and roads, many women may take long to transfer and so that window would really be useful and appropriate for us here",*

added a Senior Midwife with 8 years of experience.

**4) Provision for a section to capture key clinical notes and labour outcome data on the reverse side of the LCG A4 paper.** All interviewed HCPs reported that the WHO LCG needed to be more inclusive, and comprehensive to be more useful and appropriate in routine Ugandan settings. To be more comprehensive, HCPs reported and suggested a need to include vital clinical and labour outcomes on the LCG's reverse side of the same form to facilitate easy, quick auditing, accountability, reference, responsibility, interaction and immediate postpartum care for the mother and the newborn. According to the HCPs, inclusion of this information on two pages of one A4 paper could also mitigate the extra costs on stationary that often poses as a major challenge at basic emergency obstetric and newborn care centers. The HCPs also reported that the comprehensive data recorded on this reverse side of the A4 LCG paper in one place, could facilitate accurate, appropriate and prompt decision during and after active labour monitoring. HCPs further noted that with these considerations, the LCG had potential to facilitate team work, and make workload easier, if it could become the only required labour monitoring and management tool on ward, avoiding the challenge of what was termed as over-documentation in different places, including patient files/notes, treatment forms, fluid balance and temperature charts, something they hoped would motivate and encourage tool completion and utilization. The suggested additions according to the HCPs could include space for; 1) recording the number of antenatal care visits including date of first antenatal care visit, 2) prompting HCP to investigate or add the last hemoglobin level taken during ANC or labour, 3) any additional clinical notes not captured within the 7 sections of the LCG, e.g. use of herbal medicine during labour, disability, abnormalities, other non-labour-related diagnoses or assessments and observations, 4) outcomes of labour, e.g. date, time and type of delivery, duration of first and second stages of labour, amount of blood loss, placental examination, episiotomy/tears and repairs given if any, baby outcomes e.g., status (alive/dead), Apgar score, sex, birth weight, abnormalities, medicines given to baby including tetracycline, vitamin K, nevirapine syrup for HIV exposed newborns and cord care, 5) recording the name and cadre for the HCP that conducted the delivery and their assistant, 6) documenting immediate postpartum care given, record for postdelivery blood pressure, pulse, temperature, initiation of breastfeeding within 1 hour, new born temperature, and immunizations given prior to discharge (see details in Fig 4). All these additions, that were reported to be part of the routine care in Uganda, were suggested to make the modified LCG for Uganda a "one stop, one A4-size paper reference tool" that was comprehensive to make labour monitoring and management easier and effective. According to a 46-year-old senior midwife, who doubles as a maternity unit in charge,

*"The clinical outcomes and key notes can be added on the reverse side of this LCG to cut back on stationery and over filling and recording the same things in different places. We can then have everything we need in one place, which can trigger a quick plan for action by the whole*

*team on duty by just running through one form. . .This makes life easier and workload lighter without using more stationery and forms that is often costly and unavailable".*

*"Of course, it goes without saying that the LCG that includes space to document important clinical and labour outcomes could be not only a good strategy but a one stop center with all the reference material we need. This could potentially be what we needed all this long to pro-mote accountability and sense of responsibility since initials of persons are also captured. . .It really leaves everyone on the team with no excuse, plus, the stationery, filing and other issues would be minimized I think", added a Consultant Obstetrician with over 20 years' experience.*

### Modified LCG pilot testing

A total of 40 HCPs from MRRH approved and liked the new and modified LCG, finding it appealing, welcome, easy to use and useful to themselves and other HCPs (Table 3). All HCPs also liked the new LCG and would recommend it to others for use in labour monitoring. All interviewed HCPs also found it appropriate and implementable in our setting. All HCPs pre-ferred the new LCG for recording labour progress and majority took less than 2 minutes to completely record, plot or fill observations on the LCG after each labour assessment.

## Discussion

The enhancement of better quality, evidence based and respectful care during labour and childbirth requires concerted efforts towards better maternal and child health outcomes. This study sought to utilize mixed methods with iterative tool development approaches [23] to refine, customize and modify the new WHO Labour Care Guide to a locally contextualized acceptable tool that is useable within Ugandan and other similar settings. The simple custom-ized modifications suggested on the new WHO LCG included; 1) adding key socio-demo-graphic data compatible with existing programs to aid better planning and risk management, 2) re-ordering observations to facilitate an easy-to-use interface, flow, familiarization, engage-ment and clarity, 3) modification of key records and medication dosage to suit local context, for example, modifications to accommodate the oxytocin dose in International Units (IU) administered in a 500mL unit of crystalloids (normal saline or ringers' lactate) locally available in Uganda, including the lower limit of the normal fetal heart rate from 110 to 120 beats per minute to provide a safety margin for referral since 55% of deliveries occur at BeMONC sites (HCIIs and IIIs) that are unable to conduct caesarean section when required [40], and 4) inclu-sion of a section on the LCG reverse side to capture clinical notes, labour outcome data and other key cues to action such as triple elimination codes, Hemoglobin level and others to facili-tate auditing, accountability, reference, responsibility, interaction, team work, utilization and immediate care.

Findings from this study further showed that HCPs liked the modified WHO LCG describ-ing it as simple, easy-to-use, and quick to fill/plot compared to the old partograph. HCPs fur-ther exhibited high enthusiasm to use the LCG describing it as detailed, comprehensive, and customized to meet HCP user needs, with not only potential to reduce over documentation, save time and reduce workload but also required minimal additional training to use effectively with familiar indices. HCPs also presented the modified LCG as having potential to allow a one stop quick reference for improved clinical decisions, promote interaction, responsibility, accountability, team work and confidence among HCPs, women and others on the care team. HCPs further reported that the modified WHO LCG was acceptable, appropriate and took shorter time of less than 2 minutes to complete plotting and filling labour observations after

**Table 3. Acceptability of the modified WHO labour care guide prototype 3.**

| Variable (N = 40) | | Description | Frequency (n (%) |
|---|---|---|---|
| Acceptability of the new LCG | New LCG meets my approval | Completely disagree/disagree/Neutral | 0 (0.0) |
| | | Agree | 2 (5.0) |
| | | Completely agree | 38 (95.0) |
| | New LCG is appealing to me | Completely disagree/disagree/Neutral | 0 (0.0) |
| | | Agree | 1 (2.5) |
| | | Completely agree | 39 (97.5) |
| | I like the new LCG | Completely disagree/disagree/Neutral | 0 (0.0) |
| | | Agree | 3(7.5) |
| | | Completely agree | 37 (92.5) |
| | I welcome the new LCG | Completely disagree/disagree/Neutral | 0 (0.0) |
| | | Agree | 1(2.5) |
| | | Completely agree | 39 (97.5) |
| | Compared to the partograph, how did you find this new WHO LCG | Very/somehow bothersome | 0 (0.0) |
| | | Easy to use | 2 (5.0) |
| | | Very easy to use | 38 (95.0) |
| | How useful do you think the new LCG is to you | Not at all/somewhat useful/Neutral | 0 (0.0) |
| | | Useful | 6 (15.0) |
| | | Very useful | 34 (85.0) |
| | How useful do you think the new LCG is to other clinicians | Not at all/somewhat useful/Neutral | 0 (0.0) |
| | | Useful | 4 (10.0) |
| | | Very useful | 36 (90.0) |
| | Recommending use to others | I definitely would | 40 (100.0) |
| | | I wouldn't care one way or the other | 0 (0.0) |
| | | I definitely would not | 0 (0.0) |
| Intervention appropriateness | The new LCG seems to fit well with my work needs | Completely disagree/Disagree | 0 (0.0) |
| | | Neutral | 0 (0.0) |
| | | Completely agree/Agree | 40 (100.0) |
| | The new LCG seems suitable | Completely disagree/Disagree/Neutral | 0 (0.0) |
| | | Agree | 0 (0.0) |
| | | Completely agree | 40 (100.0) |
| | The new LCG seems applicable | Completely disagree/Disagree/Neutral | 0 (0.0) |
| | | Agree | 4 (10.0) |
| | | Completely agree | 36 (90.0) |
| | The new LCG seems like a good match | Completely disagree/Disagree/Neutral | 0 (0.0) |
| | | Agree | 0 (0.0) |
| | | Completely agree | 40 (100.0) |
| Feasibility of LCG | The new LCG seems implementable | Completely disagree/Disagree/Neutral | 0 (0.0) |
| | | Agree | 0 (0.0) |
| | | Completely agree | 40 (100.0) |
| | The new LCG seems possible | Completely disagree/Disagree/ Neutral | 1 (2.5) |
| | | Agree | 2 (5.0) |
| | | Completely agree/ | 37 (92.5) |
| | The new LCG seems doable | Completely disagree/Disagree/Neutral | 0 (0.0) |
| | | Agree | 2 (5.0) |
| | | Completely agree | 38 (95.0) |
| Would you still prefer to record labour progress info on partograph? | | Yes | 0 (0.0) |
| | | No | 40 (100.0) |

(*Continued*)

**Table 3.** (Continued)

| Variable (N = 40) | Description | Frequency (n (%) |
|---|---|---|
| **How long did it take you to plot/fill observations on the LCG after assessment** | >2 minutes | 18 (45.0) |
| | 2–5 minutes | 15 (37.5) |
| | >5 minutes | 7 (17.5) |

each assessment, and would recommend it to other clinicians for use in labour monitoring. Involving end users in development of new tools and interventions has been documented to incorporate user needs, enhance ease of use, easy uptake and sustained utilization to maximize impact [25, 26]. Other scholars have documented shorter time of active phase of labour, and reduction in Caesarean delivery rates among women monitored using a WHO LCG at a busy tertiary care institute in North India, with LCG acceptability and satisfaction levels reported to be high [17]. Another study involving a total of 136 doctors, midwives, and nurses in twelve health facilities across Kenya, Tanzania, India, Nigeria, Malawi and Argentina that applied the LCG in managing labour and childbirth of 1,226 low-risk women reported high satisfaction, usability and acceptability levels [18]. There has not been any previous study reporting time taken to completely fill labour observations on the LCG.

The utilization of user-centered approaches has potential to facilitate optimum intervention exposure, delivery, responsiveness and sustained use to improve outcomes [24]. The goal of this approach was to make the modified labour care guide easy to use, inclusive, educative, supportive, with periodic engaging prompts and cues to aid action in real time [26]. Our findings demonstrate that the involvement of HCPs in developing and or refining tools/interventions takes care of user needs, user feedback, making the interventions/tools acceptable and uptake easier. Scholars have therefore indicated the need for implementers to involve intended users during development, and introduce new interventions and approaches for use after carefully considering the context within which the intervention will be implemented to facilitate sustained integration, uptake, use and better health outcomes [7, 41].

We refined the new WHO LCG to include all the needed parameters for labour care monitoring in Uganda and improve maternal-foetal clinical and labour outcomes. The refined prototype aimed at facilitating timely auditing, accountability, reference, planning and immediate postpartum care without making it complicated to fill. Given the high patient doctor ratio in low resource settings like Uganda where the number of patients is higher compared to the available obstetric care providers which increases workload [42], such an intervention that takes a shorter time to fill could potentially enhance effective labour monitoring and better health outcomes. Indeed, many scholars [43–49] have shown that one of the challenges that hinder the utilization of the partograph is its complexity, subjectivity, which take a lot of time to fill and interpret alongside other required and parallel patient records amidst shortage of staff in health facilities that unnecessarily increases the workload of the already overwhelmed HCPs. The ability of the refined LCG prototype to be more inclusive, comprehensive and easy to use makes it a one stop user friendly tool that could facilitate easy/ quick auditing, accountability, reference to make accurate, appropriate and prompt decision during and after active labour monitoring, while mitigating the extra cost on stationary, over filing, and over documentation that could pose a major challenge especially at basic emergency obstetric and newborn care centers with limited workforce and stationery. This could enable HCPs take considerably shorter time to fill out and facilitate prompt action during labour management.

Our study attempted to customize and tailor this promising labour monitoring tool to fit within the local settings and enhance usability and acceptability. The involvement of end users to solicit their ideas, user perspectives, experiences and feedback during the development of this new tool was very crucial in ensuring that the final prototype meets their expectations and improve their overall user experience. These systematically participatory design approaches have been observed to make end users feel that their voices are heard and that their concerns have been catered for during the design, thus ultimately improving uptake and health outcomes [50]. Other studies have shown that interventions that are simply replicated in other settings may yield suboptimal outcomes compared to those locally contextualized and adapted to attain a good fit between intervention and context [51]. Therefore, characterization and refining the WHO LCG may save unnecessary time and money by addressing the potential user challenges, needs, preferences and issues early on to build trust, improve acceptability, uptake, scale up and integration in routine care by the HCPs in Uganda and similar settings.

Our study presents a number of strengths. It is nested in the principles of participatory design that emphasizes the involvement of key end user stakeholders in intervention development of a user-friendly, ccontext-specific tool aimed at supporting HCPs to effectively monitor labour in South Western Uganda. This approach could potentially facilitate acceptability, functionality, usability as well as reinforce ownership, inclusiveness, comprehensiveness, engagement and uptake among the targeted end user HCPs in Uganda and similar settings, subject to routine challenges of stationery and limited health work force. Secondly, this is the first study that reports the customization, tailoring, contextualization and time taken to completely fill the new WHO labour care guide for Uganda and similar settings, and grounded in evidence-based conceptual frameworks, making our findings meaningful, generalizable and grounded. The great enthusiasm expressed by all the HCPs presents new opportunities to actually use the tool long-term to improve intrapartum and postpartum care for women. Our study also recognises some limitations. We did not evaluate the feasibility, rate of completeness, incremental cost, sustainability and effectiveness of this new prototype to improve maternal-foetal clinical and labour outcomes. For this particular study, we assessed experiences of LCG amongst HCPs, but did not assess the labour and delivery experiences of women. Evaluation of our final prototype is ongoing to document effectiveness and its diagnostic validity compared to the partograph.

## Conclusion

Our study describes an iterative process for customizing, tailoring, refining and pilot testing the novel WHO LCG tool for labour monitoring among HCPs end users in southwestern Uganda. Involvement of targeted end-users in refining the WHO LCG tool was observed to improve acceptability, ownership, inclusiveness, comprehensiveness, engagement and uptake. The modified LCG for Uganda was found to be simple, appropriate, easy-to-use, with all HCPs preferring the new LCG for the opportunities it presented to capture all vital labour parameters and labour outcomes on a one-A4 pager, to ease accountability, reference, responsibility, interaction, utilization, teamwork and immediate care. The modified LCG was described as detailed, comprehensive and customized to meet local context HCP user needs, and majority of HCPs reported to have taken less than 2 minutes to completely record all labour observations on the LCG after each labour assessment. Further research to evaluate the modified LCG prototype's large-scale feasibility, acceptability and effectiveness in helping HCPs timely detect deviations from normal labour and prevent complications compared to the traditional partograph in different facilities and or settings is warranted.

## Supporting information

**S1 Text. Sample transcripts.**
(DOCX)

## Author Contributions

**Conceptualization:** Godfrey R. Mugyenyi, Josaphat K. Byamugisha, Esther C. Atukunda, Yarine T. Fajardo.

**Data curation:** Godfrey R. Mugyenyi, Wilson Tumuhimbise.

**Formal analysis:** Godfrey R. Mugyenyi, Wilson Tumuhimbise, Esther C. Atukunda.

**Methodology:** Godfrey R. Mugyenyi, Josaphat K. Byamugisha, Esther C. Atukunda, Yarine T. Fajardo.

**Project administration:** Godfrey R. Mugyenyi.

**Resources:** Godfrey R. Mugyenyi.

**Supervision:** Josaphat K. Byamugisha, Wilson Tumuhimbise, Esther C. Atukunda, Yarine T. Fajardo.

**Writing – original draft:** Godfrey R. Mugyenyi.

**Writing – review & editing:** Josaphat K. Byamugisha, Wilson Tumuhimbise, Esther C. Atukunda, Yarine T. Fajardo.

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
