## [Decision Letter · Decision Letter 0]

9 Feb 2024

PGPH-D-23-02500

Customization and acceptability of the WHO Labor care guide to improve labor monitoring among health workers in Uganda. An Iterative Development Study

Dear Dr. Mugyenyi,

Thank you for submitting your manuscript to PLOS Global Public Health. After careful consideration, we feel that it has merit but does not fully meet PLOS Global Public Health’s publication criteria as it currently stands. Therefore, we invite you to submit a revised version of the manuscript that addresses the points raised during the review process.

Two reviewers have provided constructive feedback.  In addition to the points raised, please also ensure that that manuscript adheres to reporting standards for the relevant study design (https://www.equator-network.org/).

We look forward to receiving your revised manuscript.

Kind regards,

Hannah Tappis, DrPH, MPH

Academic Editor

Journal Requirements:

Additional Editor Comments (if provided):

Reviewers' comments:

Reviewer's Responses to Questions

**Comments to the Author**

1. Does this manuscript meet PLOS Global Public Health’s publication criteria? Is the manuscript technically sound, and do the data support the conclusions? The manuscript must describe methodologically and ethically rigorous research with conclusions that are appropriately drawn based on the data presented.

Reviewer #1: Partly

Reviewer #2: Partly

2. Has the statistical analysis been performed appropriately and rigorously?

Reviewer #1: Yes

Reviewer #2: N/A

3. Have the authors made all data underlying the findings in their manuscript fully available (please refer to the Data Availability Statement at the start of the manuscript PDF file)?

Reviewer #1: Yes

Reviewer #2: Yes

4. Is the manuscript presented in an intelligible fashion and written in standard English?

Reviewer #1: Yes

Reviewer #2: Yes

5. Review Comments to the Author

Reviewer #1: General observation and concern

The study purports to make over 50 modifications to a WHO document. A good justification for such a major modification to a WHO document is needed. The another’s should justify the value this would make considering that such documents from WHO are done which is usually evidence based. They should indicate also if there is need for the WHO to overwhelm such a document. The paper also needs a lot of focus.

Background: This need to be beefed up to include critical literature such as: Adoption of the the WHO labor care guide in other settings. How much customization has been done in other settings and what has been the outcomes of such customization if any. Studies to evaluate such modified LCGs though RCTs or pilot should be cited. The gaps that informed the current study is are also missing.

Study design

This section needs focus and quite a bit of detail. The design itself is not stated anywhere in the paper although it looks like a mixed method design. What is stated as a design is a rumbling narrative which sometimes looks like literature review and may not be replicable.

The tool mentioned after deployment of prototype 2 should include how long it was tool used, by whom and to deliver how many women and how many facilities were involved.

A critical component of the LCG is labor and delivery experience of women? How this was assessed is not indicated.

In all the narrative in study design the sampling and proportionate allocation of HCWs at different data collection points is lacking considering that they have different characteristics.

The same is lacking at the stakeholder level considering that not all persons attending the 2 mentioned conferences could be interviewed. Also, a statement “The final feedback was obtained following a comprehensive presentation” is given but it is not clear who gave this final feedback after the use of prototype 3 and how the participants were sampled among the users from MRRH.

The tools that we used to collect data to assess critical components at each step needs to be mentioned. Data collection is just mentioned in passing in a line. This needs proper elaboration for duplicity. The study setting is too long and should be summarized

The show of “prototype development and pilot testing” needs to be clear and is it should be clear how pilot testing was done and what happened next.

It should be indicated how many stakeholders were interviewed and how were they allocated to each cadre who then used the LCG. It should also be clear which were the 4 facilities that this was done and how were they chosen.

In the HCP exit Interviews and FGDs word used here such as “At least” , “of different cadres” and 10 HCPs of “all cadres” are too general and unspecific. It is also good to indicate the timeline.

Results

This needs a lot of focus. The current result narrative looks like discussion with even a

citation “31”. Most of the results in some narrative lacks data to support and most are supported by only 1 quote from the qualitative aspect.

Before table 2 it would be good to have a table that shows the output of what lead to the changes that were made and shown in table 2.

In the segments “Customizing LCG by adding key socio-demographic data compatible with existing programs” the narrative cannot be supported by only 1 quote. There should be some results shown in a graphical or tabular form.

The many modifications done in “Modification of key fields to suit local context” should really be justified as a WHO document has gone though a lot of iterations and uses best science. Also, what appears here is actually discussion and not results. The issue of discussion happens in most other sections. The input that led to all the changes mentioned here should be very clear.

The same lack of data and narrative looking more of discussion is seen in “Inclusion of key clinical notes and labor outcome data” A table 4 is also mentioned but is not seen in the document.

Table 3 is poorly done with many cells that are having zero frequencies (0%). A bar graph would be a better way of presenting such results. Also, the narrative is more of a discussion than results presentation.

Discussion and conclusion: Since the results are not clearly presented, the discussion is speaking to results that cannot be verified as in paragraph 2&3.

Comparisons of results should be with similar studies showing similar assessed variables unlike what study cited as 34 is where authors are discussion time of filling the LCG and cites a study on outcomes.

Paragraph 3 is mainly speaking of data that is not presented. It would have been good to have utilization data.

What is in paragraph 4 is more of methodology and should be taken to the relevant section. It is not clear how the variables that are mentioned in the last sentence were evaluated and what the results were.

Paragraph 5 has also a lot of repetitions and methods. With the data presented the conclusions that the modified LCG is “simple, appropriate, easy-to-use, promote HCP-labor companion-patient interaction, responsibility, accountability, team work and confidence among HCPs” are not supported.

Reviewer #2: Review Customization and acceptability of the WHO Labor care guide to improve labor monitoring among health workers in Uganda. An Iterative Development Study

• The study presents the results of original research.

yes

• Results reported have not been published elsewhere.

yes

• Experiments, statistics, and other analyses are performed to a high technical standard and are described in sufficient detail.

Regarding data reporting, I am missing information about the following aspects:

- Knowledge of participants about the researcher/their relationship.

- Details about the study design, including the method of approach, sample size, and non-participation.

- Information about the interview guide?

• Conclusions are presented in an appropriate fashion and are supported by the data

Conclusions are presented in an appropriate fashion and are supported by the data. However, some conclusions are unexpected. For instance, sentences 438-439 lack corresponding information in the paper. Similarly, sentences 497-498 also lack supporting information in the paper. Could you consider rephrasing sentences 449-450 to enhance clarity and relevance?

• The article is presented in an intelligible fashion and is written in standard English.

Generally yes, please check some minor grammatical errors

• The research meets all applicable standards for the ethics of experimentation and research integrity.

yes

• The article adheres to appropriate reporting guidelines and community standards for data availability.

See above about point of reporting guidelines.

Other points:

- Did you also left investigate any points for improvement during the exit interviews?

- What changes did you make in point 5: presentation of LCG and its protocols/guidelines in attractive colors, size, shape and content?

- In your results section, can you elaborate on the percentages of HCP that liked/agreed to the new adapted LCG?

- Can you elaborate on the fact why you had so many outstanding results regarding the updated LCG?

- Please check refs 7. Looking at abstract, doesn’t seem to correspond?

- Please check refs 23-26 (don’t seem to fit with the text) (143)

- Please be consistent in terminology prototype 1 versus prototype one (I prefer the first version) 166/107

- Can you explain /rephrase the part about baseline fetal heart rate (285).

- Can you elaborate on the reactions/results of local practitioners about the use of Shared Decision making in these settings? Do you have any information about this available? (250)

- I still don’t fully understand why you decided to skip the ‘ other medications’ section (307) in your labor guide. What about other ‘labor- relevant medications’ such as Magnesiumsulfate etc.

6. PLOS authors have the option to publish the peer review history of their article (what does this mean?). If published, this will include your full peer review and any attached files.

**Do you want your identity to be public for this peer review?** For information about this choice, including consent withdrawal, please see our Privacy Policy.

Reviewer #1: No

Reviewer #2: No

---

## [Editor Report · Decision Letter 1]

12 Apr 2024

Customization and acceptability of the WHO Labor care guide to improve labor monitoring among health workers in Uganda. An iterative development, mixed method study

PGPH-D-23-02500R1

Dear Dr Mugyenyi,

We are pleased to inform you that your manuscript 'Customization and acceptability of the WHO Labor care guide to improve labor monitoring among health workers in Uganda. An iterative development, mixed method study' has been provisionally accepted for publication in PLOS Global Public Health.

Best regards,

Hannah Tappis, DrPH, MPH

Academic Editor